# Comprehensive Assessment of the Universal Healthcare System in Dentistry Japan: A Retrospective Observational Study

**DOI:** 10.3390/healthcare10112173

**Published:** 2022-10-30

**Authors:** Shy Chwen Ni, Carlos Thomas, Yu Yonezawa, Yasushi Hojo, Takehiko Nakamura, Kenichiro Kobayashi, Hiroki Sato, John D. Da Silva, Takuya Kobayashi, Shigemi Ishikawa-Nagai

**Affiliations:** 1Department of Oral Medicine, Infection and Immunity, Harvard School of Dental Medicine, Boston, MA 02115, USA; 2Harvard Kennedy School, Cambridge, MA 02138, USA; 3School of Dental Medicine, Iwate Medical University, Morioka 020-8505, Japan

**Keywords:** universal healthcare system, dentistry, health policy, expenditure, aging society

## Abstract

Japan’s universal healthcare insurance is facing economic challenges due to the advanced aging society, however, objective data of dental expenditure has never been introduced. This study aimed to identify the associated factors with dental expenditures using government-provided digitized insurance claims data and calculated the spending in the context of dental cost per person (DCPP). Seven associated factors analyzed were age, demographic, geographic, socioeconomic, regional wealth, the impact of the 8020-national campaign implementation (keep 20 teeth at age 80), and the effect of the home-visit dentistry for the elders. The average DCPP was high in older populations (75+) in all prefectures. The prefectures with the highest and lowest DCPP were significant compared to other states and retained their respective places in the cost hierarchy over the four years. The prefectures with more citizens participating in government assistance programs (GAP) had greater DCPPs. Dental costs were significantly related to geographic regions, age, per capita income, government assistance program prevalence, office complete denture frequency, and home visit care per patient. With a growing aging population, dental care costs will continue to increase, burdening its fiscal future. Associated factors identified should be considered to control the contentious increase of healthcare cost.

## 1. Introduction

Japan’s universal health system was established in 1961 and has been commended for its accomplishment in good population health at a low cost while maintaining an equitable system. Since 1986, Japan has had the world’s highest life expectancy. The government has made a continuous effort to allow all Japanese citizens to have insurance according to employment status, residence, and age and have affordable access to health services. Everyone of all ages gets insured through the employee-based plan, a community-based plan, or the late-stage medical care plan. There are uniform fee schedules across the nation, which allows equity and containment of costs and access for everyone to comprehensive dental care. The merit of Japanese social insurance is that dental health service is included in the general health service by law and covers more than half of dental costs for the citizens. Japan reported having the highest number of dental visits among the countries in the Organization for Economic Co-operation and Development (OECD), with nearly half the level of edentulism for its seniors compared to several European countries [1].

Oral health is often overlooked as a public matter in many countries, resulting in no or minimal covered dental services in the medical health plan. Japan addressed this issue in the early development of its health system. In 1989, “8020 (Eighty-Twenty) campaign” was promoted by the Ministry of Health and Welfare (MHLW) and the Japanese Dental Association, which aimed to help their citizens to retain “20” or more natural teeth when they reach “80” years old and over. The motion was based on supporting evidence that a minimum of 20 natural teeth is necessary for adequate chewing function, nutritional intake, and social well-being [2]. Since then, the government has subsidized many initiatives through collaborations with local health authorities and dental associations and contributed to the marked improvement in the oral health and quality of life of older people. As a result, the proportion of the elderly aged 80 years and above who have at least 20 teeth has increased from 40.2% (2011) to 51.2% (2016), reported in a recent MHLW national survey [3]. 

Japan has become an advanced-aging society where one in 10 people is 85 years old or older. Disease structure changes while associated health expenses increase in growing elders who are disabled, live alone or have both conditions. In 2000, the Japanese government launched the long-term care insurance system (LTCI), anticipating the demographic change and national finances concerning population aging and related healthcare needs [4]. In 2006, the government introduced the disability prevention program by arranging social activities for older adults and helped reduce disability incidents [4]. The provision of LTCI promotes community-based integrated care, which provides qualified insurers with in-home services, day/short stay services, special nursing homes, and other services [5]. In-home services include dental management and treatments via portable equipment. Domiciliary dental care for the elderly improved their oral health and enjoyment of food, enhancing their self-esteem and social acceptance [2]. 

Though Japan has successfully maintained equality and promoted health, evaluating the system’s sustainability should be prioritized now because long-term care costs and premiums are expected to increase in the future. Half of the LTCI budget comes from taxes derived from the national government (25%), prefectures (12.5%), and municipalities (12.5%), while the other half of the finance comes from premiums paid by every citizen aged 40 years or over. The long-term care cost has increased from USD 32.7 billion (in the year 2000) to USD 106.4 billion (in the year 2019), and this number is projected to exceed USD 136.4 billion by the year 2025 [4]. Furthermore, qualified older adults who use the LTCI services are only required to pay a 10% copayment, while the remaining 90% is covered by the LTCI budget [6]. Therefore, the financial burden relies heavily on the shrinking working population and the government [7,8]. 

In 2011, some authors of the Lancet series reviewed the progress of the Japanese health system in detail on aspects of evolution, achievements, and challenges and provided constructive feedback. They recommended that the accomplishment of Japan’s premier health should be exemplary to the world; however, Japan’s engagement with global health has not been outstanding relative to its substantial potential. Attributing factors may be government fragmentation, a weak civil society, and a lack of transparency and assessment [9]. Nevertheless, in recent years, Japan has prioritized global health in its international diplomacy by conveying a strong political message at several conferences to advocate for global health and universal health coverage. Furthermore, as a host at the 2016 summit of the Group of Seven (G7), Japan pointed out the need to communicate more widely with the global community in an era of globalization [10]. 

To facilitate transparency and assessment, the Japanese government has implemented a standardized infrastructure for health information. All digitized claims data on anonymized insured individuals are gathered by MHLW and registered in National Receipt Database (NRD). Since 2011, these data have been accessible to third parties, researchers, and local governments, to help conduct health service research and create data-driven health policies [11]. Our research team was approved to obtain access to assess the database provided by the MHLW and dental health expenditures using universal health insurance from 2015 onward were analyzed. 

Reports on dental health expenditure regardless public or private insurances are scarcely find in the literature due to limited data source and outcome interpretation with inferences. Surveys were commonly used in the previous studies, which limited to respondent’s records and lacked generalizability [12,13,14]. This study uses national claims data to examine the dental health expenditure of the universal health system in Japan, which provides a more objective and comprehensive report. 

Based on Japan’s dental delivery system, the association between the dental cost per person and its associated factors in age, demographics, socioeconomics, the progress of the 8020-national campaign, and home-visit dentistry as a special care system for the elderly population were analyzed. We hypothesized that the dental cost per person is positively associated with increasing age, use of services by the elderly, and geographic differences. 

## 2. Materials and Methods

### 2.1. Study Design Using Open Data for Research Use 

This retrospective descriptive study was approved by Japan’s MHLW in 2018 and obtained Institutional Review Board of Iwate Medical University, Japan (IMU_01301). Figure 1 represents the scheme of the study design. The dataset from 2015 to 2018 was retrieved from the NRD, where providers submitted medical-dental fee bills to a payment examination organization. The MHLW then collects the data, anonymizes it, and creates the database [15]. The dental data became publicly available in 2015 as the “open data, “ aggregating the dataset annually based on clinical diagnosis by sex, five-year age groups, and insurance claims [16]. The proposed national data was extracted in the summer of 2019 (Figure 1). The dataset was obtained from all dentists/dental practices that registered in the Japanese universal healthcare system. The sample size of this study is equivalent to the total claim number submitted in the universal healthcare system, and therefore, study bias is virtually nonexistence.

### 2.2. Indicator of Dental Health Expenditure 

In Japan, the health expenditure in dental clinics is 6.8% of the total healthcare expenditure [6]. Datasets on the dental cost per person per month (DCPP) were obtained and analyzed at the national and sub-national levels. DCPP is the average dental cost covered by universal health insurance per patient per month. This variable was calculated by dividing the total dental expenditures by the corresponding population. The DCPP was used as an indicator for a measure of dental health expenditure for descriptive analysis and graphical representations, stratifying by factors related to demographic, geographic, socioeconomic, and regional wealth [12]. 

### 2.3. Study Variables 

#### 2.3.1. Geographic

Japan consists of 47 prefectures, and each prefecture has a local welfare bureau. It was reported that regional differences were observed in terms of the prevalence of caries, dentist ratio per 100,000 people, and economic status [5]. The highest dentist-to-population ratio in Tokyo was double that of the Fukui Prefecture in 2016. Therefore, the DCPP of all prefectures is of interest to evaluate whether the distribution of dentists influences dental expenditures [17]. 

#### 2.3.2. Socioeconomic Characteristics 

A decline in the use of dental services is commonly associated with a decline in a state’s wealth. A prefecture’s relative wealth can be estimated by gross domestic product (GDP) and average per capita income [18]. On the other end of the spectrum, to measure wealth, the percentage of people in a prefecture that receive the government assistance program (GAP) was utilized. According to law, public assistance for the poor is a responsibility of the state. The proof of the economic condition “poverty” is the sole criteria for receiving assistance in cases when the minimum cost of living exceeds the final income. Types of assistance include livelihood, housing, education, medical, and long-term care, etc. [19]. Thus, a state’s wealth based on the following factors: GDP, per capita income, and the government assistance program were analyzed to understand whether there is an association with the DCPP.

#### 2.3.3. Age Levels 

As people age, more health services are utilized, including dental services. Four age groups (0–14, 15–64, 65–74, and >75) were evaluated on their DCPP. Age 0–14 is considered the youngster group. Age 15–64 is viewed as the working group. The older age group is subdivided into 65–74 years old or over 75. These groupings with different age levels are due to the difference in copayments for dental services. Those over 75 and children under six are responsible for 10% of copayments, those aged 70–74 pay 20%, and insurers in other age ranges pay 30%. In addition, the retirement age is approximately 65 years old in Japan. Since there is different pay responsibility, it would be worthwhile to know if there is a difference in the DCPP in the retirement group that pays 20% copayments and compares it to the stratified elderly group that pays 10% copayments.

#### 2.3.4. Dental Service Relevant to the Elderly 

8020 Campaign

The main goal of this campaign is to encourage Japanese citizens to retain 20 teeth as they reach 80 years old. Complete denture (CD) and removable partial dentures (RPD) are the most demonstrative in the datasets as a measure for loss of teeth. There are different types of claim codes of RPD. The numeric number after the RPD code represents the number of teeth replaced and fabricated for the RPD. This means that RPD 9~11, RPD 5~8, and RPD < 4 have 20 or more remaining teeth if calculated from 32 teeth for complete arches. The frequency of the CD claims per dental practice is also analyzed in all prefectures and compared with the DCPP in different age groups. 

Home Visit Dentistry (HVD)

There has been an increase in home care services, especially among the elderly, by 37.9% from 2015 to 2019, reflecting the increased needs due to disability [16]. Those who cannot visit the dental office and live in nursing homes or long-term care facilities rely on dental practitioners for on-site services. The government provides reimbursement for home visit fees per visit, which are set based on the number of patients seen in a building. For example, USD 110 is reimbursed when only one elderly/patient is seen in a building (Type I). USD 36 is reimbursed for each 2nd to 9th elderly/patient seen in a building (Type II). USD 18 for the number 10th or above elderly/patients seen in a building (Type III). The frequency and cost of each home visit type were evaluated and associated with dental costs per person.

### 2.4. Statistical Analysis

The spearman’s correlation coefficient was used to analyze the association between DCPP and the different variables. One-way ANOVA was used to compare the significant difference of DCPP among prefectures, years, age groups, and frequency and cost for HVD. Then, Tukey HSD test and Bonferroni test were used for post hoc analysis. Multiple linear regression analysis was used to analyze association between DCPP and GAP, per capita income, HVD, state GDP, and dentists density.

## 3. Results

The summary of total data extracted from the national database was shown in Table 1. The total number of data point for DCPP was 564 (47 states times 12 months) for each year, in total 2256. Total claim point was extracted in 4 age groups for each year. The number of claims for dentures and HVD were also extracted.

### 3.1. National and Prefecture Level Dental Cost

The national average monthly dental cost per person (DCPP) showed a slight downward trend from 2015 to 2018, demonstrated in Figure 2a. Furthermore, the DCPP of all 47 prefectures was calculated, with six prefectures being outliers. The prefecture with the greatest DCPP was Hokkaido (labeled ① in Figure 2b,c). Hokkaido’s DCPP was 40% greater than Mie, the prefecture with the least DCPP (labeled in Figure 2b,c). The DCPP of Hokkaido (largest) was significantly higher than all other 46 states in all four years, and Aomori and Akita (2nd and 3rd largest) were higher than the other 42 states. In contrast, the DCPP of Mie was significantly lower than the other 42 states. The bottom three states are from the same local welfare bureaus (Tokaihokurkku-Kinki), and the top three are in two northern bureaus (Hokkaido-Tohoku).

### 3.2. Demographic Characteristic of Dental Cost 

The association of the DCPP with the age groups (0–14, 15–64, 65–74, and 75+) was observed. As predicted, with aging, people are spending more on dental services (Figure 3a). DCPP rose significantly in increased age groups, and the trend remained the same in all four years. The 75+ age group is responsible for a greater share of the average DCPP (Figure 3b) and the only age group that showed a static and persistent slight increase in each subsequent year (18.1% in 2015, 18.7% in 2016, 19.5% in 2017, and 20.1% in 2018).

### 3.3. Prefecture Wealth and Dental Cost

The DCPP was compared to population dentistry and dental clinic density to explain the influence of regional health infrastructures. Both were negatively correlated with DCPP, but neither was significant (Table 1). DCPP negatively correlates with a prefecture’s per capita income (Figure 4b). As the per capita income of the prefecture increases, there is a corresponding decrease in DCPP. Additionally, the DCPP positively correlates with a prefecture’s government assistance program (GAP) enrollment rate. The greater percentage of a prefecture’s population in GAP, the higher the corresponding DCPP (Table 2 and Figure 4a). The capital prefecture, Tokyo, is an outlier in this analysis. 

There was a weak moderate negative correlation between DCPP and DDS density, state GDP, and per capita income. In contrast, there was positive correlation with GAP and HVD (Spearman’s correlation coefficient, Table 2 and Table 3). 

After controlling for each covariate, per capita income remained significantly inversely correlated with DCPP in all four years from 2015–2018. A significant positive correlation was found between the GAP and DCPP, and the HDV and DCPP was found only in year 2015, and 2018, respectively. 

### 3.4. Elderly Dental Services Utilization and Dental Costs

Over the years, the frequency of all denture claims decreased, which demonstrated an increase in retained teeth. The types of the prosthesis with the most declined use were CD and RPD with greater than 12 denture teeth and between 9~11 denture teeth (Figure 5a). As a result, the expenditure on dentures decreased from 1.128 billion to 1.022 billion dollars which amounted to a reduction of 106 million dollars. 49.9% and 10.2% of the savings are attributable to decreased CDs and RPDs for more than 12 teeth, respectively (Figure 5b). All 47 prefectures reported decreasing trends for frequency of complete dentures claims per dental practice (FCDPP) and 85% (40 prefectures) being significant (Figure 5c). 

Further analysis of the association of the DCPP with FCDPP in 2018 showed a strong positive correlation for the age 65–74 group (R = +0.555). Data is shown on the moderate correlation for the 75 and over group (R = +0.45). The same trends were observed in 2015–2017. Overall, the denture claims have decreased, but the prefecture with higher FCDPP is still consistent with a higher DCPP. The number of home visits to dentistry (HVD) of all types increased over the four years, along with the associated expenditure (Figure 6a,b). The frequency of monthly HVD per patient was also positively correlated with the DCPP with a significance (R = +0.319, Figure 6c). 

## 4. Discussion 

Japan is an interesting case study for multiple reasons. As a world economic leader—the third-largest economy by gross domestic product—it is also a leader in health delivery—among the few universal health care countries that include dentistry as an essential benefit [20,21]. To the best of our knowledge, this is the first work in Japan to comprehensively analyze per capita dental costs across factors including age, income, and dental services. Other recent studies that also evaluated the dental cost were US, Canada, and Brazil. The US study stratified age 65+ from the entire population and found that average dental expenditure per person increased by 27% for the overall population and 59% for the elderly population in the past two decades [12]. The Canadian study examined the association of dental service utilization and per capita public dental care spending in different jurisdiction and found that higher spending and greater population coverage increases dental visits [13]. The Brazil study used 2.5% of private insurance reported data on household expenditure per capita and found that higher education level and income were associated with higher spending [14]. Our study showed similar findings and with more credible sources of national data claims that include the entirety of population. 

This study aimed to understand better the current state of dental delivery expenditures in Japan and the strides made by 8020 campaign. Many studies have been published analyzing the effects 8020 has had on masticatory function, body mass index (BMI), and heart disease [22]. However, there is limited literature relating the 8020 campaign to factors such as expenditures and denture claims. Analyzing both complete and partial dentures provide a new measure to evaluate the 8020 campaigns. The fewer complete dentures and removable partial dentures (RPDs) with fewer teeth are claimed, the more successful the campaign outcome is because patients keep their teeth longer and do not require more substantial oral prosthetics later in life [23]. Home visit dentistry allows the elderly to access dental care without traveling and has become popular in recent years. Most of the patients are older than 70 years old; many of whom receive dental home visits live in group homes and are more likely to be exempt from out-of-pocket payments [24]. Denture repairs and fabrication are the most common treatment modalities for this patient population [25]. As such, the goal was to report on relevant trends for home visit dentistry in the context of dental costs per person. Other research has corroborated our research that HVDs have become increasingly integrated, especially for the elderly. One study found that 8% of homebound adults received home dental visits. Most of them were aged 81–90, and one-third lived in a group home. Additionally, 10% of those receiving this benefit were exempt from out-of-pocket payments [24]. In our study, significant increase of frequency and cost of HVD was observed and positively correlated with DCPP. These findings demonstrate that HVD is more needed for the aging society.

To control prices in the healthcare industry, the central government sets nationally uniform fee schedules every two years and enforces regulations for insurers and healthcare workers, which has allowed Japan to contain its healthcare expenditures and spend, on average less than its fellow OECD countries [3,26]. However, Japan’s healthcare spending has been gradually rising. It is now the fifth-greatest healthcare spending country as a proportion of GDP: 11% compared to the 8.8% average among other OECD countries [27]. Japan has been facing economic stagnation coupled with an aging society consuming more expensive care. The International Monetary Fund published that Japan’s healthcare spending has greatly outpaced economic growth since the 1990s. This is largely a function of the increasing share of its retired population. For example, among OECD members, Japan is well above the 90th percentile in per capita healthcare spending and per capita long-term care spending for aged over 65 patients [28]. 

The other aspect of this research shows the strides Japan’s 8020 campaign has made since its implementation in decreasing complete and extensive partial dentures. However, universal health insurance related to dental coverage can still be improved upon. The current health insurance system only covers treatments for existing diseases such as fillings, endodontic treatment, crowns, bridges, dentures, and extractions and interventions focused on treatments [5]. While the number of decay teeth has decreased, the number of sound (untreated and healthy) teeth for adults has also decreased, and the number of filled teeth has increased [5]. Additionally, the participation rates in periodontal disease examinations remain low [29]. Japan’s dental insurance, which is comprehensive by global metrics, excludes prevention services; if Japan is to secure the public health of its population, preventative services will be integral. 

Japan’s oral health infrastructure is impressive. Patients do not have to choose between medical or dental services—their copayment percentages are the same [11]. The fees are similar across providers, and relevant government insurance schemes contract with many practitioners to deploy dental services, including treating patients in long-term facilities. However, as previously studied, patient income is a strong predictor of seeking dental care [30,31]. Additionally, as found by this study, Japan’s costs are significantly related to a region’s wealth, population age, percentage of people using government assistance programs, and the number of complete dentures done in practice.

This study found a significant negative correlation between DCPP and per capita income, which implies those with higher income received less dental care coverage from the universal healthcare system. This can also be interpreted that those with higher income choosing non-insured dental treatments, such as ceramic restorations, dental implants, and metal-based dentures. This finding suggests that policymakers consider expanding insured dental care and services. Like most US private insurance systems, the procedure-specific co-payment rate could be one solution to balancing dental expenditure and expanding insured dental care and services. DCPP also has a significant association among the states and local welfare bureaus. Though there is no clear explanation for this phenomenon, the standards of insurance coverage approval among the prefectures and bureau may differ, and it might have an effect on DCPP. 

One of the limitations of this study is the years of database obtained. Although the Japanese government implemented a new digitalization system of the national database open to the third party for research in 2011, the complete database for dentistry was stored from 2015. Therefore, this study still provides significant insight regarding the universal healthcare system usage trend. It is important to note that the presented data trend in this study may differ dramatically during the 2019–2022 pandemic, and further investigation is warranted. All the data obtained from the government used in this study is not readily available to the public. Therefore, our findings can be useful for all journal readers and the general population in Japan.

## 5. Conclusions

The dental expenditure–DCPP is significantly related to geographic regions, age, per capita income, government assistance program prevalence, denture frequency, and home visit care by the patient. All these factors concern a country where 40 percent of the population is expected to be 65 years or older soon. Japan may need to consider health care service financing before the burden of caring for its citizens becomes too great.

## Figures and Tables

**Figure 1 healthcare-10-02173-f001:**
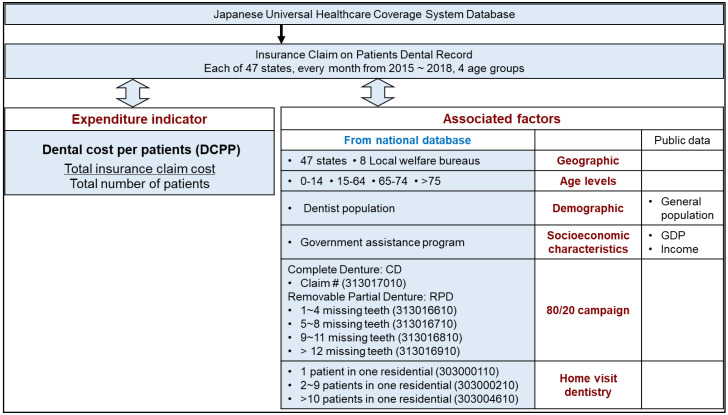
Scheme of the study design. Study data were extracted from the Japanese Universal Healthcare System (UHCS) database. Monthly Dental Cost Per Patient (DCPP) was calculated and the association between DCPP and associated factors of six categories were analyzed.

**Figure 2 healthcare-10-02173-f002:**
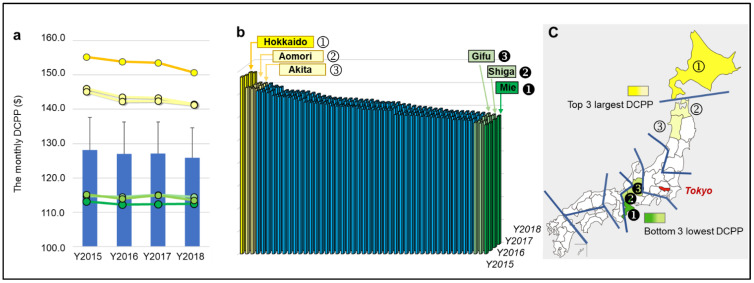
Association between geographic factors and monthly DCPP (dental cost per person). Yellow indicates the top 3 state with higher DCPP, and green indicates the lowest 3 states. (**a**): Monthly average DCPP of 47 states. There is a slight decrease in the monthly average DCPP of 47 states over this time frame, but no statistical significance was observed in this decline. (**b**): Average monthly DCPP in all 47 states. The DCPP of Hokkaido (Top 1) was significantly higher than all other 46 states (*p* < 0.01, One-way ANOVA and Tukey HSD test), and Aomori and Akita were higher than the other 42 states (*p* < 0.01). In contrast, the DCPP of Mie was significantly lower than the other 42 states (*p* < 0.01). (**c**): The bottom three states are in the same local welfare bureaus (Tokaihokuriku-Kinki), and the top three are in the north two bureaus (Hokkaido-Tohoku).

**Figure 3 healthcare-10-02173-f003:**
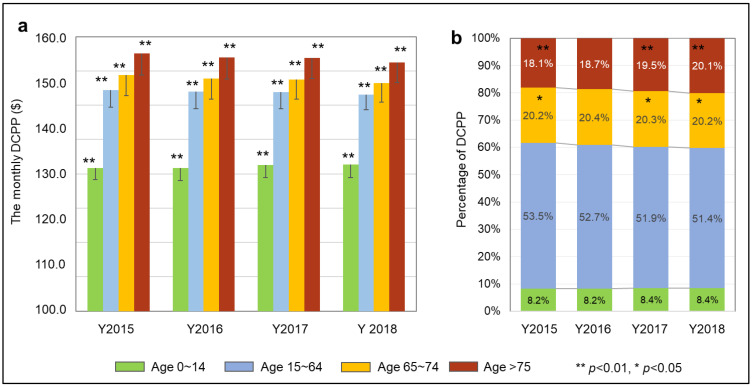
Association between age levels and DCPP. (**a**): Average monthly DCPP in 4 age groups. A statistically significant difference (*p* < 0.01 **, One-way ANOVA and Tukey HSD test) in DCPP among the age groups was observed, showing that as people age, they spend more on dental services. (**b**): Proportion of DCPP in 4 age groups. The 2017 and 2018 monthly DCPP proportion for the 75+ group was significantly greater than in 2015 (*p* < 0.01 **).

**Figure 4 healthcare-10-02173-f004:**
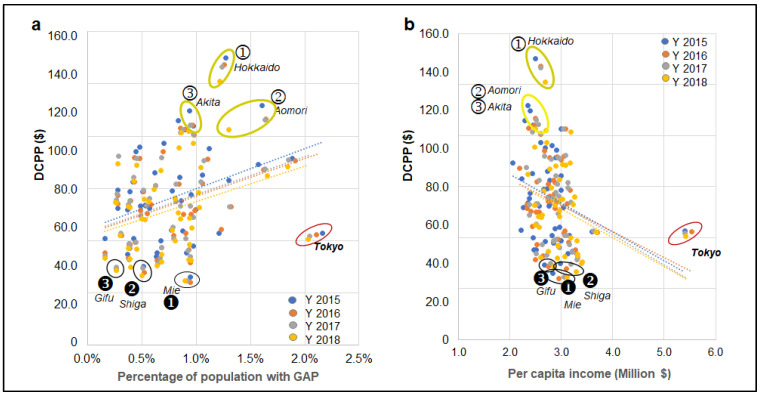
Association between DCPP and demographic and socioeconomic factors. (**a**): Hokkaido (①) and Aomori (②) have a greater population receiving GAP (government assistance program) than the national average (>1 SD), and Gifu (❸) and Shiga (❷) indicated a lower percentage (<1 SD). (**b**): In contrast, Hokkaido (①), Aomori (②), and Akita (③) indicate lower per capita income than the national average. The capital city, Tokyo, was an outlier.

**Figure 5 healthcare-10-02173-f005:**
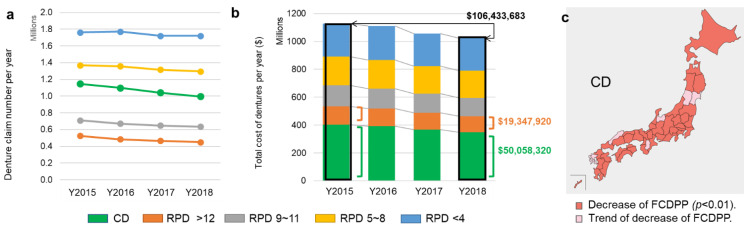
The trend of frequency of dentures under the 80/20 national campaign. (**a**): The number of dentures claims and expenditure from 2015 to 2018 in 5 types of dentures. The number of denture claims decreased over the year (R: −0.99, *p* < 0.01, Spearman’s correlation coefficient) on CD, RPD > 12, and RPD 9–11 teeth. (**b**): Total cost of dentures from 2015 to 2018. The cost of dentures decreased from 1.128 to 1.022 billion dollars. A significant decrease (R= −0.99, *p* < 0.01) was observed on CD and RPD for 12–14 teeth (respectively contributing to 49.9% and 10.2% decreased cost of $106,433,683, the sum of all four years). (**c**): The frequency of CD claims per dental practice (FCDPP) in each of the 47 states from 2015 to 2018. All states showed a trend of decrease, and 85% of states (40 states) had a statistically significant decline in use (*p* < 0.01).

**Figure 6 healthcare-10-02173-f006:**
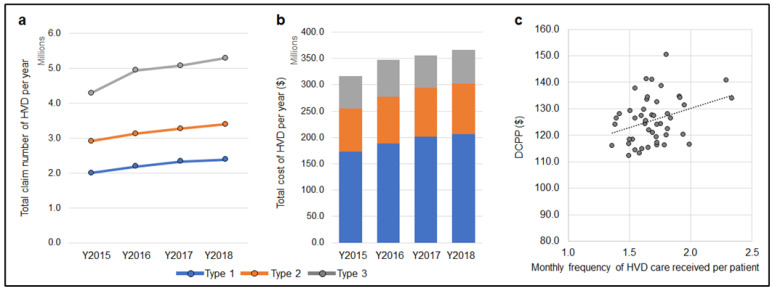
Frequency of HVD (Home Visit Dentistry) and its association with DCPP. (**a**): The frequency of HVD type 1, 2, and 3 increased significantly during the four years (*p* < 0.01, One-way ANOVA and Tukey HSD test). (**b**): The expenditure of HVD type 1, 2, and 3 increased significantly (*p* < 0.01). (**c**): There was a moderate positive correlation between DCPP and monthly frequency of HDV care received per patient (R = +0.319, *p* < 0.01, Spearman’s correlation coefficient).

**Table 1 healthcare-10-02173-t001:** Summary of numbers of data extracted.

	Year	Y2015	Y2016	Y2017	Y2018
		47 states × 12 months			
Total number of data for DCPP	2256 in total	564	564	564	564
Number of registered dentist		68,592	68,737	68,940	68,609
Total claim point	Age 0–14	17,727,415,626	21,601,273,240	22,613,241,915	23,171,906,152
Number of patients		20,585,083	24,880,345	25,402,529	25,905,229
Total claim point	Age15–64	115,017,660,176	138,025,379,467	138,965,364,030	140,784,309,556
Number of patients		89,580,930	107,948,845	108,979,236	111,316,374
Total claim point	Age 65–74	41,533,423,936	50,996,245,433	51,363,162,948	51,565,234,545
Number of patients		30,852,430	38,284,019	38,770,916	39,436,842
Total claim point	Age > 75	36,908,447,828	47,003,037,455	50,055,080,950	52,930,763,759
Number of patients		25,058,846	32,254,026	34,484,550	37,001,263
80/20 campaign/Number of total denture claim		5,512,140	5,379,173	5,189,849	5,097,025
CD		1,145,702	1,097,686	1,040,780	996,884
RPD for >12 missing teeth		524,798	482,777	464,221	450,077
RPD for 9~11 missing teeth		710,221	671,261	647,773	634,643
RPD for 5~8 missing teeth		1,369,572	1,357,342	1,315,941	1,294,740
RPD for 1~4 missing teeth		1,761,847	1,770,107	1,721,134	1,720,681
Number of total HVD claim		9,191,078	10,240,233	10,673,841	11,059,549
Type 1		1,997,558	2,182,137	2,332,446	2,382,789
Type 2		2,910,276	3,121,972	3,269,609	3,392,247
Type 3		4,283,245	4,936,124	5,071,786	5,284,513

**Table 2 healthcare-10-02173-t002:** Association between demographic and socioeconomic characteristics and DCPP.

Indicators		Demographic (Population Density)	Socioeconomic Characteristics			Special Support for Elderly
		DDS Density	State GDP	Per Capita Income	GAP (Government Assistance Program)	HVD (Home Visit Dentistry)
DCPP	2014 2015 2016 2017	R: −0.0943 R: −0.1588 R: −0.1902 R: −0.1926	R: −0.2716 R: −0.1829 R: −0.1892 R: −0.1992	R: −0.3471 R: −0.3262 R: −0.3210 R: −0.3535	R: +0.3675 R: +0.3347 R: +0.3150 R: +0.2934	R: +0.2323 R: +0.2785 R: +0.2999 R: +0.2827

**Table 3 healthcare-10-02173-t003:** Multiple linear regression analysis on DCPP controlling for DDS density, state GDP, per capita income, GAP and HVD.

	2015				2016				2017				2018			
	Regression Coefficient	*p*-Value	Lower 95%	Higher 95%	Regression Coefficient	*p*-Value	Lower 95%	Higher 95%	Regression Coefficient	*p*-Value	Lower 95%	Higher 95%	Regression Coefficient	*p*-Value	Lower 9 95%	Higher 95%
DDS Density	0.369	0.619	−1.117	1.856	0.218	0.757	−1.198	1.634	0.233	0.740	−1.177	1.643	0.235	0.717	−1.066	1.537
State GDP	1.033×10^5^	0.558	−2.502 × 10^5^	4.569 × 10^5^	1.268 × 10^5^	0.463	−2.190 × 10^5^	4.726 × 10^5^	1.226 × 10^−5^	0.453	−2.040 × 10^−5^	4.492 × 10^−5^	1.450 × 10^−5^	0.316	−1.436 × 10^−5^	4.336 × 10^−5^
Per Capita Income	−0.001	* 0.049	−0.002	−3.591 × 10^−6^	−0.001	* 0.047	−0.002	−1.2584 × 10^5^	−0.001	* 0.034	−0.002	0.000	−0.001	* 0.012	−0.002	0.000
GAP	6.661 × 10^4^	* 0.041	2.953 × 10^3^	1.303 × 10^5^	5.581 × 10^4^	0.074	−5.622 × 10^3^	1.172 × 10^5^	5.472 × 10^4^	0.076	−6024.959	1.155 × 10^5^	4.598 × 10^4^	0.134	−1.477 × 10^4^	1.067 × 10^5^
HVD	1.443 × 10^3^	0.059	−56.534	2.942 × 10^3^	1355.632	0.083	−185.672	2896.936	1517.015	0.069	−121.808	3155.838	1628.257	* 0.037	106.753	3149.762

## Data Availability

The data presented in this study are available on request from the corresponding author. The data are not publicly available due to the government policy in Japan.

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
