# Peer review of "Comprehensive Assessment of the Universal Healthcare System in Dentistry Japan: A Retrospective Observational Study"

_healthcare, 2022, doi:10.3390/healthcare10112173_

Round 1

Reviewer 1 Report

The paper shows that dental costs were associated with some factors in NDB dataset of Japan. It is an interesting study. However, there are some issues. The paper needs to be revised.

1) The authors should follow the STROBE statement and add a checklist.

2) The number of data is unclear, because there is no flowchart in the result section.

3) It is unclear what the authors add in this study.

4) There are no multivariate analyses.

5) The authors do not consider the changes in cost and system during the period, which is a big issue.

Introduction

1) There are no previous studies and the past findings about the association between dental costs/care costs and other related factors. Which one is unknown or knownPlease add appropriate references despite with/without universal health system and totally revise the part.

2) Why did the authors use USD? The topic is about Japan and journal locates in Switzerland. In the materials and methods, they use Yen, but, in the results section, they use USD. Please unify the unit.

3) The percentage of copayment has been changed during the period; i.e., 10%-30%. Please add the details and refer the changes in the analyses.

4) Please add a hypothesis.

Materials & Methods

1) Please add some explanation for 313017010 and more in the Figure 1.

2) Please add a paragraph for statistical parts. All tests should be described. Furthermore, Bonferroni correction is required.

3) Some Arabic numbers are at the beginning of a sentence (L193, 194).

4) Please add the detail information about changes in costs and copayment, and how to adjust them. Please re-analyze the data.

5) Please add more information because there are no important parts following the STROBE checklist.

6) Why did the authors use the category of partial denture? Please add more detail in the section.

7) The number of teeth present is required in this paper. Please add the data.

Results

1) Please add a flowchart.

2) Please revise the cut-off of P values, such as 0.05/47 in each figure.

3) Please add the name of statistical analyses in all figures and number of sample/participant.

4) In the Figure 3, 5, 6 and Table1, how many samples do the authors use? Please add the details as above.

5) Please add the results of multivariate analyses.

6) What is a main outcome? It is unclear.

Discussion

1) Please add comments about generalizability and limitation.

2) Please discuss the association between dental care cost and other factors with appropriate references regardless of no health care coverage of dental care in other countries.

3) Please revise the conclusion based on only the results in this study.

Author Response

Thank you for your feedback, we all appreciate your guidance.

We revised the manuscript according to your feedback and suggestions. 

Sincerely

Reviewer 2 Report

Thank you for the opportunity to review the manuscript entitled ”Comprehensive assessment of the universal healthcare system in dentistry Ja-pan: a retrospective observational study”. The data is interesting and the manuscript is very well written. 

I only felt to have more explanations regarding these differences between regions and more detailed intepretation in the Discussion section.

I recommend the authors to take this into consideration, it would be nice for the Readers who are not so familiarized with Japanese medical system to have a more reflected image.

FOr example, even from the begining of the manuscript, the authors sustained that ”Dental costs were significantly related to geographic 23 regions, age, per capita income, government assistance program prevalence, office complete den- 24 ture frequency, and home visit care per patient. ”. It would be nice that these results to be explained and interpreted in Discussion section.

This is my only comment, the manuscript nis well-written and results are clearly presented.

Author Response

Thank you for your feedback, and we all appreciate your guidance.

We revised the manuscript according to your feedback and suggestions.

Sincerely

Reviewer 3 Report

The study is very interesting and genuine. However, the authors should address the following points to improve the quality of the manuscript:

- The term "super-aging society" in the abstract should be replaced by an alternate scientific term.

- The authors should include a short sentence in the abstract about the outstanding research gap and question (please keep the word limit).

- It is advisable to use passive voice in scientific writing.

- Please add the null hypothesis/hypotheses to the end of the introduction section.

- The collected data are related to the period of 2016-2018. The authors should justify why the data is 4-6 years old and how is it still applicable to date?

- Figure 2 was made of multiple graphs, please consider splitting in different figure for clarity and readability.

- The authors should add the study limitations and future directions for research to the discussion section.

- The conclusion may be summarized and outlined in bullets.

Author Response

(The authors gave the same response as above.)

Round 2

Reviewer 1 Report

The paper was overall improved. However, there is a typo (L391-L473).

Author Response

Thank you for your guidance.

We carefully checked the spell and revised it.

Sincerely

Reviewer 3 Report

The authors have done all the necessary changes

and the paper can be accepted in the present 

form.

Author Response

Thank you for your guidance.

Sincerely